

# Decoherence and relaxation of topological states in extended quantum Ising models

**Hannes Weisbrich[1], Wolfgang Belzig[1] and Gianluca Rastelli[1,2,★]**

**1** Fachbereich Physik, Universität Konstanz, D-78457 Konstanz, Germany
**2** Zukunftskolleg, Universität Konstanz, D-78457, Konstanz, Germany

★ gianluca.rastelli@uni-konstanz.de

## Abstract

We study the decoherence and the relaxation dynamics of topological states in an extended class of quantum Ising chains which can present a multidimensional ground state subspace. The leading interaction of the spins with the environment is assumed to be the local fluctuations of the transverse magnetic field. By deriving the Lindblad equation using the many-body states, we investigate the relation between decoherence, energy relaxation and topology. In particular, in the topological phase and at low temperature, we analyze the dephasing rates between the different degenerate ground states.

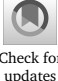
# 1   Introduction

The quantum Ising chains introduced in quantum magnetism [1–6] represent a class of exactly solvable many-body systems [7] that exemplifies one-dimensional quantum phase transitions [8–12]. More recently, the quantum Ising model was studied in the non-equilibrium regime to investigate the dynamical behavior of quantum phase transitions, e.g. the quenching in a driven Ising chain [13–18], the Kibble-Zurek mechanism [19,20], the Loschmidt echo of a single impurity coupled to the Ising chain [21], the engineered quantum transfer [22], the quantum superposition of topological defects [23], the decoherence dynamics in the strong coupling regime [24] as well as the role of quantum correlations in quantum phase transitions [25–27]. Importantly, the generalized class of Ising models can be characterized by a topological number [28–32] and, in the topologically nontrivial phase, localized states can occur at the end of an open chain [1,4] or at the interface separating regions with different topological number [33]. This is associated to the ground state degeneracy in the limit of long chains. For instance, in the case of the $XY$ Ising chain, these end-states correspond to the Majorana zero mode of the one-dimensional fermionic Kitaev model [34–36]. Depending on the topological number, the extended models of Ising chains can present more than two end-states, viz. several Majorana zero modes [28–32,37,38]. The topology can even change by simply adding a single impurity at one end of an open chain [39].

The quantum Ising model has been experimentally implemented using neutral atoms in optical lattices [40], lattices of trapped ions [41–43], Rydberg atoms [44], Josephson junctions [45] and superconducting qubits [46–48]. These realizations have to be considered as open quantum systems [49,50] since the degree of freedom corresponding to the spin can be readily affected by interaction with the environment. In general, the interplay between dissipation and interactions in quantum many-body systems presents a rich phenomenology [51–57]. A quantum reservoir-engineering can lead to desired quantum states [58–61]. Dynamical instabilities can occur in the phase diagram of driven dissipative systems [62–65]. The decoherence and relaxation dynamics can be characterized by a slow, algebraic decay [66] or by anomalous diffusion [67]. Other interesting effects are the formation of maximally entangled states protected against phase-flip noise [68] and the non-monotonic critical line in the phase diagram of a system with competing dissipative interactions [69].

Although, a priori, spin lattices synthesized in mesoscopic devices can encode Majorana states [70–72], which have potentially application in topological quantum computation, the dissipative interaction affecting such systems distinguish them from other realizations. In topological insulators and semiconducting nanowires, Majorana states are protected by fermion parity conservation against the dephasing induced by bosonic fluctuations. By contrast, when one transforms the Ising chains into the fermionic lattices via the Jordan-Wigner transformation, one has to transform consistently the spin operators coupled to the environment. As a consequence, the system is not anymore topologically protected and the dissipation in the transformed model can induce, for instance, inelastic transitions between states of different parity.

Remarkably, the parity still plays a crucial role when the spins have a longitudinal dissipative coupling, namely they are coupled to the environment via the same spin component coupled to the transverse magnetic field, see Fig. 1. This model of longitudinal dissipation was considered to address the quantum phase transition using the mean field approximation [73] or an exact approach based on path integral [74]. This kind of dissipation was also analyzed to investigate dynamical phase transitions [75–78], non-equilibrium states in presence of temperature differences [79], quantum diffusion [80] and relaxation dynamics in the strong coupling limit [24].

For a single spin/qubit, the dissipation in the Markovian regime can be characterized by

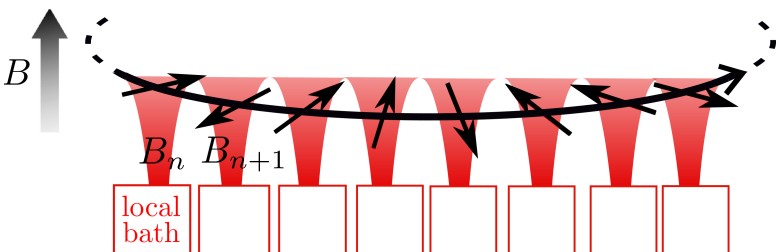

Figure 1: Schematic figure of the interacting spin chain with local coupling to the environment. Each spin is coupled to a local bath via the spin component parallel to the magnetic field of the Ising model.

two time constants: the energy relaxation time $T_1$ (the characteristic time scale in which the spin releases energy to the environment) and the dephasing time $T_2$ (the time scale after which a coherent superposition of two quantum states reduces to a statistical mixture). In this work, we analyze the energy relaxation and the decoherence dynamics for an extended class of quantum Ising chains formed by $N$ spins under the effect of longitudinal dissipative interaction. This represents the natural dephasing mechanism for the spins in absence of interaction. Such regime can occur when, for instance, the individual energy relaxation time of the single qubit $T_1$, is much larger than the individual dephasing time $T_2$, with $T_1$ and $T_2$ defined in absence of interactions. In general, one has $T_2 < T_1$ and, in some cases, one can also approach $T_2 \ll T_1$, e.g. for flux and fluxonium qubits [81]. Assuming this regime, we consider only the longitudinal coupling as the dominant interaction with the surrounding environment. In order to show that the topological protection is conserved in presence of this longitudinal dissipative interaction, we derive the appropriate Lindblad equation for the many-body system.

In the limit of low temperature, we investigate the correlations between the topology of the spin chain, characterized by a winding number $g$, and the decoherence in the multidimensional ground state subspace. For the simplest case of the transverse Ising model ($g = 1$), we discuss the crossover from the trivial to the topological phase. We distinguish different contributions, of thermal or topological origin, appearing in the dephasing rate for an initial state given by a coherent superposition of the ground state and the zero-energy excitation. In the topological regime, these two states are almost degenerate for $N \gg 1$ and the decoherence rate is set by the overlap of the square modulus of the wave functions of the state localized at the left and the right end of the chain (one Majorana zero mode). This term decreases exponentially by increasing the chain length $N$ such that the coherent superposition survives for a long time, viz. the transient regime to achieve statistical mixture of the two ground states is long compared to the other time scales of the system.

We generalize the results of the simple transverse Ising model by studying an extended model which includes three body, next nearest neighbor interaction, with $g = 2$ in the topological phase. In this case the ground state subspace is fourfold degenerate, with two ground states in each parity sector (even and odd) and with two zero modes whose wave functions are localized at the ends of the chain. Hence, the extended model opens the possibility to implement adiabatic quantum computation in a given parity ground state subspace. Naturally, this is impossible in the simple transverse Ising model, as the ground states have different parity, which is conserved by the Hamiltonian. Within each parity subspace of the extended model and at low temperature, we find a formula for the decoherence rate that is proportional to a generalized overlap of wave functions involving both Majorana zero modes. This generalized overlap factor still decreases exponentially with the length of the chain in the limit $N \gg 1$. Finally, in the extended model, the lowest energy excitations in each parity subspace can relax

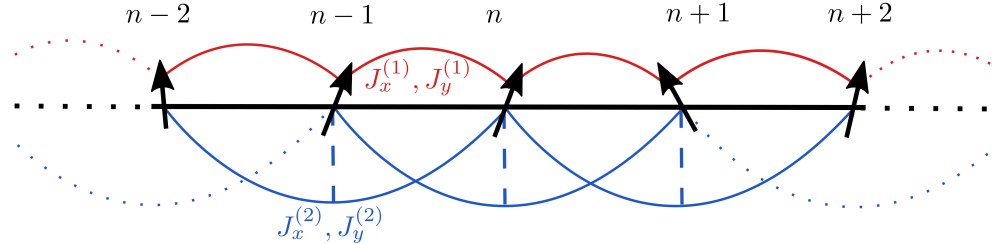

Figure 2: Chain of interacting spins. The nearest neighbor interaction couples two neighboring spins in the $x$- and $y$-component with coupling constants $J_x^{(1)}, J_y^{(1)}$ (red lines). The three-body interaction (blue lines) with coupling constants $J_x^{(2)}$ and $J_y^{(2)}$ involves the $x$- and $y$- components of two next nearest neighbor spins at position $n-1$ and $n+1$, and the $z$-component of the intermediate spin at position $n$.

towards one of the two possible ground states. By preparing the system in one of these excited states, we study the decay rates and the final probability of occupation of the different ground states in a (long) transient regime. We show that the latter quantity is associated to the behavior of the wave functions of the Majorana zero modes and of the single particle spectrum in the topological region $g = 2$.

This work is organized as follows. In Sec. 2 we recall the class of exactly solvable extended Ising models and their topological characterization. Using the Jordan-Wigner transformation, we map the spin model to the fermions model which we diagonalize using the generalized Bogoliubov transformations. In Sec. 3 we discuss the interaction with the local baths and we derive a Lindblad equation starting from the Bogoliubov operators. The Lindblad operators are associated to the transitions between different many-body states of the system. In Sec. 4 we show the results for the transverse Ising model. Afterwards, in Sec. 5 we discuss the results for the simplest extended quantum Ising model characterized by a fourfold ground state degeneracy for which we analyze the dephasing dynamics of the two ground states of same parity. We summarize our results in Sec. 6.

## 2 Extended quantum Ising models and topology

In this section we introduce an extended class of quantum Ising models. We set the notation $\hbar = k_B = 1$. These models describe one-dimensional interacting spin chains with equal spins and nearest neighbor interaction as well as next nearest neighbor interaction as shown in Fig. 2. We consider chains of finite length with $N$ spins [82]. The Hamiltonian of these chains reads

$$H_S = H_c + \epsilon H_{b.c.}, \tag{1}$$

with

$$H_c = -B \sum_{n=1}^{N} \sigma_n^z - \sum_{n=1}^{N-1} \left( J_x^{(1)} \sigma_n^x \sigma_{n+1}^x + J_y^{(1)} \sigma_n^y \sigma_{n+1}^y \right)$$
$$- \sum_{n=2}^{N-1} \left( J_x^{(2)} \sigma_{n-1}^x \sigma_n^z \sigma_{n+1}^x + J_y^{(2)} \sigma_{n-1}^y \sigma_n^z \sigma_{n+1}^y \right), \tag{2}$$

and the boundary term

$$H_{b.c.} = -J_x^{(1)}\sigma_N^x\sigma_1^x - J_x^{(2)}\left(\sigma_{N-1}^x\sigma_N^z\sigma_1^x + \sigma_N^x\sigma_1^z\sigma_2^x\right)$$
$$- J_y^{(1)}\sigma_N^y\sigma_1^y - J_y^{(2)}\left(\sigma_{N-1}^y\sigma_N^z\sigma_1^y + \sigma_N^y\sigma_1^z\sigma_2^y\right), \tag{3}$$

with the spin operators $\sigma_n^\alpha$ at site $n$ with $\alpha = x, y, z$ obeying the algebra $[\sigma_n^\alpha, \sigma_m^\beta] = \delta_{n,m} i\epsilon_{\alpha\beta\gamma}\sigma_n^\gamma$ and $\epsilon_{\alpha\beta\gamma}$ the Levi-Civita symbol. There are two different boundary conditions: $\epsilon = 1$ for the closed chain and $\epsilon = 0$ for the open chain. The model of Eqs. (1-3) consists of an external transverse magnetic field $B$ in the $z$-direction, a pairwise nearest neighbor interaction for the spins $n$ and $n+1$ in the $x$- and $y$-component of the spins, and a three-body interaction with next nearest neighbor interaction in the $x$- and $y$-component of the spins at position $n-1$ and $n+1$ mediated by the $z$-component of the intermediate spin at position $n$. A schematic picture is reported in Fig. 2. This extended interaction is chosen, as it can be projected to a simple next nearest neighbor interaction in the fermionized Hamiltonian via the Jordan-Wigner transformation defined by

$$c_n^\dagger = \nu_n\sigma_n^-, \qquad \nu_n = \prod_{m=1}^{n-1} e^{i\frac{\pi}{2}(1-\sigma_m^z)}, \tag{4}$$

with $\sigma_n^\pm = \sigma_n^x \pm i\sigma_n^y$ and the fermionic (spinless) operators satisfying the anticommutation relations $\{c_n, c_m^\dagger\} = \delta_{nm}$ and $\{c_n, c_m\} = \{c_n^\dagger, c_m^\dagger\} = 0$. Setting $J_{x\pm y}^{(1)} = J_x^{(1)} \pm J_y^{(1)}$, $J_{x\pm y}^{(2)} = J_x^{(2)} \pm J_y^{(2)}$ and the parity operator

$$P = \prod_{n=1}^{N}(1 - 2c_n^\dagger c_n), \tag{5}$$

the resulting fermionic Hamiltonian reads

$$H_c = -B\sum_{n=1}^{N}\left(1 - 2c_n^\dagger c_n\right) - \sum_{n=1}^{N-1}\left(J_{x+y}^{(1)} c_n^\dagger c_{n+1} + J_{x-y}^{(1)}(c_{n+1}c_n + h.c.)\right)$$
$$- \sum_{n=1}^{N-2}\left(J_{x+y}^{(2)} c_n^\dagger c_{n+2} + J_{x-y}^{(2)}(c_{n+2}c_n + h.c.)\right), \tag{6}$$

and

$$H_{b.c.} = P\left(J_{x+y}^{(1)} c_N^\dagger c_1 + J_{x-y}^{(1)}(c_1 c_N + h.c.) + J_{x+y}^{(2)}\left(c_{N-1}^\dagger c_1 + c_N^\dagger c_2\right)\right.$$
$$\left. + J_{x-y}^{(2)}(c_2 c_N + c_1 c_{N-1} + h.c.)\right). \tag{7}$$

Using the parity projection operators $\mathcal{P}^\pm = (1 \pm P)/2$ with $\mathcal{P}^+ + \mathcal{P}^- = \mathbb{1}$ we project the Hamiltonian onto the subspaces of even and odd parity

$$H_S = \mathcal{P}^+ H_S \mathcal{P}^+ + \mathcal{P}^- H_S \mathcal{P}^- \equiv H_S^+ + H_S^-. \tag{8}$$

For a closed chain of finite length, owing to the discrete spatial translation invariance, one defines the unitary transformation in the momentum space as $c_{k\pm} = e^{-i\pi/4}/\sqrt{N}\sum_n e^{i\pi k^\pm n/N}c_n$, whereas $k^+$ ($k^-$) takes all odd (even) integers between $-N$ and $N$ for the even (odd) subspace such that we can write the Hamiltonian (for one parity subspace) in the following form

$$H_S^\pm = 4\sum_{k^\pm}\left[\mathcal{B}_x(k^\pm)s_{k^\pm}^x + \mathcal{B}_z(k^\pm)s_{k^\pm}^z\right], \tag{9}$$

where we have introduced the pseudo spin representation

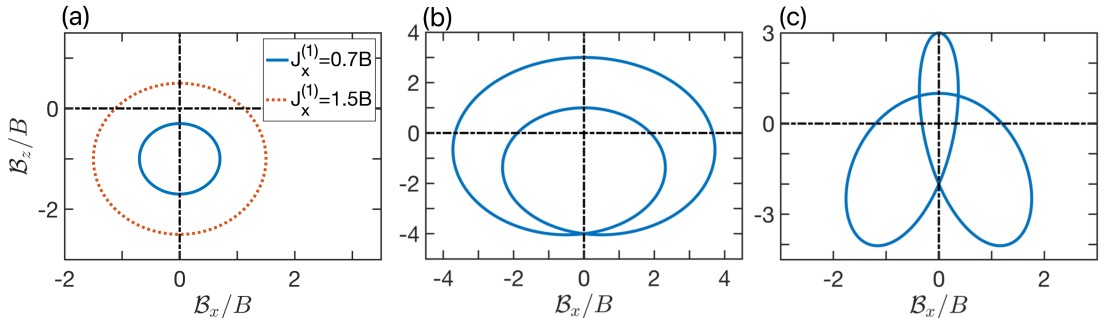

Figure 3: Example of the topological winding number in the extended Ising model with closed boundary condition. (a) The transverse Ising model with $J_y^{(1)} = J_y^{(2)} = J_x^{(2)} = 0$ has $g = 0$ for $B > J_x^{(1)}$ (trivial phase) and $g = 1$ for $B < J_x^{(1)}$ (topological phase). (b) Example of $g = 2$ with $J_x^{(1)} = 0.5B$, $J_x^{(2)} = 3B$ and $J_y^{(1)} = J_y^{(2)} = 0$ and (c) $g = 2$ with $J_x^{(1)} = J_x^{(2)} = B$, $J_y^{(1)} = 0$ and $J_y^{(2)} = 2B$.

$$s_k^- = (s_k^+)^\dagger = \hat{c}_k \hat{c}_{-k}, \quad s_k^z = \frac{1}{2}\left(\hat{c}_k^\dagger \hat{c}_k + \hat{c}_{-k}^\dagger \hat{c}_{-k} - 1\right),\tag{10}$$

with $s_k^x = (s_k^+ + s_k^-)/2$, and the effective magnetic field

$$\mathcal{B}_x(k) = J_{x-y}^{(1)} \sin(k\pi/N) + J_{x-y}^{(2)} \sin(2k\pi/N),\tag{11}$$

$$\mathcal{B}_z(k) = J_{x+y}^{(1)} \cos(k\pi/N) + J_{x+y}^{(2)} \cos(2k\pi/N) - B.\tag{12}$$

In the long chain limit $N \gg 1$, the Eqs.(11,12) describe a closed curve in the plane $\mathcal{B}_x, \mathcal{B}_y$ when one varies parametrically the wave vector $k$ in the first Brillouin zone. This curve is uniquely defined by the set of parameter in the Hamiltonian. The number of closed loops around the origin defines the winding (topological) number of the spin system [28–32]. More specifically, the winding number $g$ in the $xz$-plane is defined as the line integral on the closed curve spanned parametrically by the vector $k$

$$g = \frac{1}{2\pi} \oint_C \frac{1}{\mathcal{B}_x^2 + \mathcal{B}_z^2}(\mathcal{B}_z d\mathcal{B}_x - \mathcal{B}_x d\mathcal{B}_z),\tag{13}$$

and determines the number of clockwise rotations around the origin. Examples are given in Fig. 3. The transverse Ising model with $J_y^{(1)} = J_x^{(2)} = J_y^{(2)} = 0$ is represented by a circle with radius $J_x^{(1)}$ and the center shifted by $B$, see Fig. 3(a). Hence for $J_x^{(1)} < B$ the origin is not within the circle, thus this regime is referred to as the trivial regime with $g = 0$. For $J_x^{(1)} > B$ the origin is within the circle leading to $g = 1$, thus we are in the topological regime. The topological regime with $g = 1$ results in a twofold degenerate ground state, which is equivalent to a Majorana zero mode in the fermionic picture for the open chain [28–32]. Other examples with $g = 2$ are reported in Fig. 3(b) and Fig. 3(c). In these cases, the system has a fourfold degenerate ground state in the open chain with two Majorana zero modes [28–32].

More explicitly, the Hamiltonian of the open chain $H_c$ can be expressed in the following diagonal form

$$H_c = E_{GS} + \sum_i E_i \gamma_i^\dagger \gamma_i,\tag{14}$$

with the fermionic Bogoliubov operators $\gamma_i$ and the eigenenergies $E_i$. The Bogoliubov operators are determined by the unitary transformation which we define as

$$\gamma_i = \frac{1}{2} \sum_n \left(\left(\psi_{i,n}^L + \psi_{i,n}^R\right) c_n + \left(\psi_{i,n}^L - \psi_{i,n}^R\right) c_n^\dagger\right),\tag{15}$$

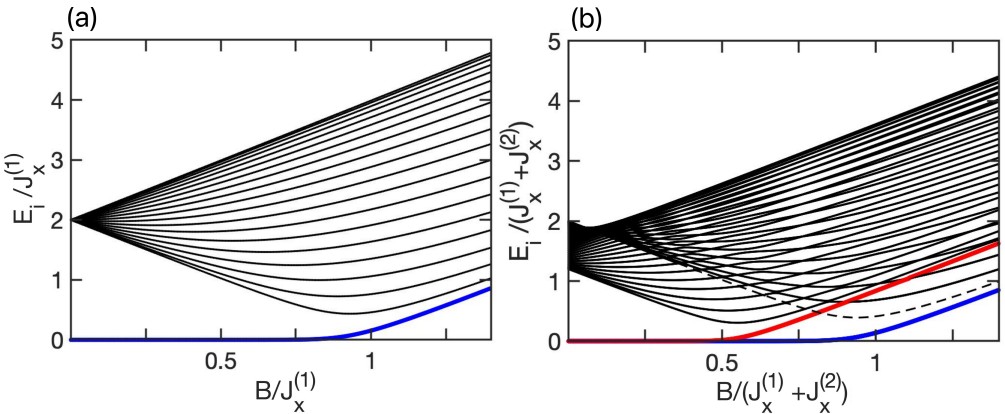

Figure 4: Single particle spectrum $E_i$ of an open chain formed by $N$ spins. (a) The transverse Ising model defined by $J_y^{(1)} = J_y^{(2)} = J_x^{(2)} = 0$ with $N = 20$ and (b) the extended model with parameters $J_x^{(2)} = 4J_x^{(1)}$, $J_y^{(1)} = J_y^{(2)} = 0$ and $N = 40$. Here the blue and red lines correspond to the two zero modes, with vanishing energy in the topological regime. The dotted line represents an effective, secondary gap $E'_{\text{gap}}$ as discussed in Sec. 5.

where the coefficients (wave functions) $\psi_{i,n}^{L/R}$ and the eigenenergies $E_i$ are determined by solving numerically the Lieb-Schultz-Mattis equations [1] (see appendix A). In some cases analytic solutions are available. For instance, in the case of the transverse Ising model, the coefficients $\psi_{i,n}^{L/R}$ read [1]

$$\psi_{i,n}^L = f_N \, \sin[\kappa_i(N + 1 - n)], \tag{16}$$

$$\psi_{i,n}^R = f_N \, \text{sign}\left(\frac{\sin(\kappa_i)}{\sin(\kappa_i N)}\right)\sin(\kappa_i n) \quad \text{(transv. Ising)}, \tag{17}$$

whereas $f_N$ is the normalization constant such that $\sum_n |\psi_{i,n}^{L/R}|^2 = 1$. The energies of the single particle spectrum are given by $E_i = 2\sqrt{B^2 + (J_x^{(1)})^2 - 2BJ_x^{(1)}\cos(\kappa_i)}$, whereas the possible $k$ values are the solutions of the following transcendental equation:

$$\tan(\kappa_i(N + 1)) = J_x^{(1)}\sin(\kappa_i)/(J_x^{(1)}\cos(\kappa_i) - B).$$

An example of the single particle spectrum $E_i$ of the open transverse Ising chain is shown in Fig. 4(a). The lowest excitation - which hereafter we denote as $i = 0$ - has imaginary solution for $\kappa_0 = iq_0$ in the topological regime ($B < J_x^{(1)}$) leading to localization of $\psi_{0,n}^L$ and $\psi_{0,n}^R$ at the ends of the chain [1] and with energy $E_0$ that vanishes $E_0 \approx 0$ in the limit $N \gg 1$. This represents the zero energy mode whose wave function, in the limit of long chain $B/J_x^{(1)} \ll N$, is given by [1]

$$\psi_{0,n}^L \simeq \sqrt{(J_x^{(1)}/B)^2 - 1}\ e^{-q_0 n} \quad \text{(transv. Ising)}, \tag{18}$$

and $\psi_{0,n}^R = \psi_{0,N+1-n}^L$, namely we have a Majorana zero mode localized at the ends of the chain with $1/q_0 \approx 1/\ln(J_x^{(1)}/B)$ as decay length. The wave functions $\psi_{0,n}^L$ and $\psi_{0,n}^R$ are plotted in Fig. 5(a). Hence the system has a twofold degenerate ground state with the ground state $|GS\rangle$ defined in the trivial regime and the zero-energy Bogoliubov excitation $\gamma_0|GS\rangle = |0\rangle$. The energy gap is given by $E_{\text{gap}} = E_0$ in the trivial regime $B > J_x^{(1)}$ which strongly decreases around $J_x^{(1)} = B$ (i.e. the critical value of the quantum phase transition in the thermodynamic limit), see Fig. 4(a). By contrast, in the topological regime $B < J_x^{(1)}$ the lowest fermionic excitation

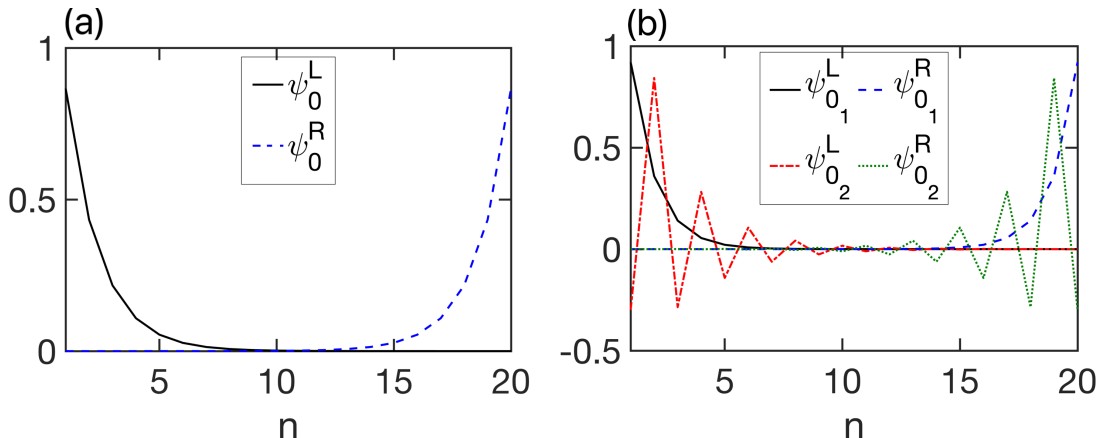

Figure 5: Behavior of the wave functions of the zero-energy end states (Majorana zero modes) for open spin chains with length $N = 20$. (a) $\psi_{0,n}^L$ and $\psi_{0,n}^R$ of the transverse Ising model in the topological regime $g = 1$ with $J_x^{(1)} = 2B$ and $J_y^{(1)} = J_y^{(2)} = J_x^{(2)} = 0$. (b) $\psi_{0_1,n}^L$, $\psi_{0_1,n}^R$, $\psi_{0_2,n}^L$, $\psi_{0_2,n}^R$ of the extended model in the topological regime $g = 2$ with $B = J_x^{(1)} = 0.25 J_x^{(2)}$ and $J_y^{(1)} = J_y^{(2)} = 0$.

$E_0$ vanishes (twofold degenerate ground state) and the gap is defined by $E_{\text{gap}} = \min_i E_i$ for $i \neq 0$.

As a second model analyzed in this work we consider the model Hamiltonian introducing an additional interacting term in the transverse Ising model by setting $J_x^{(2)} > 0$ (but still $J_y^{(1)} = J_y^{(2)} = 0$). In this case, the system can approach a winding number $g = 2$, see Fig. 3(b), in the topological regime $B < J_x^{(2)} - J_x^{(1)}$. An example of the single particle spectrum for this case is shown in Fig. 4(b) for a chain with $N = 40$ spins. Here we observe the appearance of a first zero-energy excitation, which we denote $i = 0_1$, close to the point $B = J_x^{(2)} + J_x^{(1)}$ in which $E_{0_1}$ is strongly reduced (this corresponds to the first critical point where the gap closes exactly in the thermodynamic limit). Furthermore a second zero-energy excitation, which we denote $i = 0_2$, also appears close to the point $B = J_x^{(2)} - J_x^{(1)}$ in which $E_{0_2}$ is also strongly reduced (this corresponds to the second critical point where the gap closes again in the thermodynamic limit). After this point, in the long chain limit, the ground state subspace is almost fourfold degenerate with the states $|GS\rangle$ and $\gamma_{0_1}^\dagger \gamma_{0_2}^\dagger |GS\rangle = |0_1, 0_2\rangle$ in the even parity subspace and $\gamma_{0_1}^\dagger |GS\rangle = |0_1\rangle$ and $\gamma_{0_2}^\dagger |GS\rangle = |0_2\rangle$ in the odd parity subspace. Again, $|GS\rangle$ is connected to the single ground state of the trivial regime. In the topological regime $B < J_x^{(2)} - J_x^{(1)}$ with $g = 2$, the wave functions of the two zero-energy states localized at the ends of the chain (Majorana zero modes) have the following analytic formulas [28]

$$\psi_{0_1,n}^L \simeq c_1 e^{-q_{0_1} n} \qquad (J_x^{(2)} > 0), \tag{19}$$

$$\psi_{0_2,n}^L \simeq c_2 e^{-q_{0_1} n} + c_3 e^{i\pi n} e^{-q_{0_2} n} \qquad (J_x^{(2)} > 0), \tag{20}$$

and $\psi_{0_i,n}^R = \psi_{0_i,N+1-n}^L$ (for $i = 1, 2$), whereas the coefficients $c_1, c_2, c_3$ are set by the conditions that the wave functions are normalized and orthogonal ($\sum_n \psi_{0_i,n}^{L/R} (\psi_{0_j,n}^{L/R})^* = \delta_{0_i,0_j}$). The inverse of the decay lengths associated to the pairs of localized modes are given by

$$q_{0_1} = \ln\left(\frac{2J_x^{(2)}}{\sqrt{\left(J_x^{(1)}\right)^2 + 4BJ_x^{(2)}} - J_x^{(1)}}\right), \quad q_{0_2} = \ln\left(\frac{2J_x^{(2)}}{\sqrt{\left(J_x^{(1)}\right)^2 + 4BJ_x^{(2)}} + J_x^{(1)}}\right). \tag{21}$$

An example of the wave functions $\psi_{0_1,n}^L$, $\psi_{0_1,n}^R$ and $\psi_{0_2,n}^L$, $\psi_{0_2,n}^R$ is plotted in Fig. 5(b). Notice that the wave function associated to the second zero-energy states has an oscillatory behavior with a longer decay length, whereas the first mode has a decay similar to the single zero energy mode of the transverse Ising.

## 3 Lindblad equation for the interacting chain

Before discussing the spin chain of interacting spins, we first recall the results for a single spin coupled to a thermal bath which can be expressed as

$$H_{1,\text{spin}} = -B\sigma_z + \sigma^x S_b^x + (\sigma^z - 1)S_b^z + H_{\text{bath}}, \tag{22}$$

with $S_b^\alpha$ (hermitian) operators of the bath (to simplify the notation, we neglect the $y$ component). The shifted interaction in the $z$-component, namely the term $\sigma^z - 1$, does not affect the rest of our analysis. If the bath is an ensemble of independent harmonic oscillators, this shift can be formally removed by unitary (polaron) transformation that displaces the equilibrium position of the oscillators. In the limit of weak coupling with the environment, assuming the Born-Markov approximation and the secular approximation, one can derive the Lindblad equation, see for example [50]. This means that one factorizes the density matrix of the whole system formed by the spin and the bath, with the bath at thermal equilibrium $\rho \approx \rho_s \otimes \rho_{\text{bath}}$. The relevant quantities in this approach are the Fourier transforms of the correlators of the bath operators at thermal equilibrium

$$\kappa_\alpha(\omega) = \int_{-\infty}^{\infty} dt \, e^{i\omega t} \, \text{tr}_{\text{bath}} \left[ S_b^\alpha(t) S_b^\alpha(0) \rho_{\text{bath}} \right]. \tag{23}$$

Moreover, in the Markov approximation, the memory effects are neglected assuming a fast decay of the bath correlators in comparison to the time scales of the system. In the last step the secular approximation is used, where the fast rotating terms are neglected, as they average out on larger time scales. Using this approach for the single dissipative spin one finds the energy relaxation rate of the single spin which reads $1/T_1 = \kappa_x(2B) + \kappa_x(-2B)$. The fluctuations of the longitudinal component of the bath operator leads to pure dephasing whose rate is given by

$$\frac{1}{T_\phi} = 2 \lim_{\omega \to 0^+} \kappa_z(\omega). \tag{24}$$

Hereafter we assume $1/T_\phi$, the dephasing rate of the single spin, as a given, effective parameter. The total dephasing rate is given by $1/T_2 = (1/T_\phi) + 1/(2T_1)$.

In the limit of $T_\phi \ll T_1$, the longitudinal $\sigma_z$-coupling to the bath is the dominant one. Therefore, if we regard the system at time scale smaller than $T_1$, one can neglect the transverse interaction of the qubit with the environment. Hence, we consider the pure dissipative longitudinal interaction affecting the individual spins and the total Hamiltonian reads

$$H_{tot} = H_c + \sum_n \left( \sigma_n^z - 1 \right) S_{n,b}^z + \sum_n H_{n,\text{bath}}. \tag{25}$$

We remark that, even if this kind of coupling to the environment in $\sigma_z$-direction leads to a pure dephasing in the non-interacting spin chain, this is not true anymore when we consider the interacting case. In the latter case, the appropriate basis in the perturbative scheme between system and environment are the many-body eigenstates. The latter are not eigenstates, in general, of the local spin operator $\sigma_n^z$. In other words, this interaction can also lead to energy

relaxation in the case of interacting spin chains. Note, however, that the dissipative interaction in Eq. (25) commutes with the parity operator and can not induce energy relaxation between states of different parity.

Having in mind long spin chains, we focus on the case in which the local baths are uncorrelated and homogeneous such that we can write

$$\int_{-\infty}^{\infty} dt \, e^{i\omega t} \langle S_{n,\text{b}}^z(t) S_{m,\text{b}}^z \rangle_{\text{bath}} = \delta_{n,m} \kappa(\omega), \tag{26}$$

which is a realistic assumption for a homogeneous spin chain with locally separated spins with average spacing larger than the correlation length of the fluctuations of the environment. For open chain conditions, we express the local spin operator $\sigma_n^z$ in term of the fermionic Bogoliubov operators

$$\sigma_n^z - 1 = -2 \sum_{i,j} \left[ A_{i,j,n} \gamma_i^\dagger \gamma_j + B_{i,j,n} (\gamma_i \gamma_j + \gamma_j^\dagger \gamma_i^\dagger) \right], \tag{27}$$

with

$$A_{i,j,n} = \frac{1}{2} \left( \psi_{i,n}^R \psi_{j,n}^L + \psi_{i,n}^L \psi_{j,n}^R \right), \tag{28}$$

$$B_{i,j,n} = \frac{1}{4} \left( \psi_{i,n}^L \psi_{j,n}^L + \psi_{j,n}^R \psi_{i,n}^L - \psi_{i,n}^R \psi_{j,n}^L - \psi_{j,n}^R \psi_{i,n}^R \right). \tag{29}$$

Following the standard approach similar to a single spin [50], using the Markov-Born approximation combined with the secular approximation, one can derive the Lindblad equation. In the final result, the relevant quantity are the ladder (Lindblad) operators, which can be obtained by considering the spectral decomposition of the coupling operator $\sigma_n^z - 1$ to the local bath at site $n$. In the interaction picture we write these operators as

$$e^{iH_c t} \left( \sigma_n^z - 1 \right) e^{-iH_c t} = -2 \sum_{i,j} A_{i,j,n} \gamma_i^\dagger \gamma_j e^{i(E_i - E_j)t} - 2 \sum_{i,j} B_{i,j,n} \left( \gamma_i \gamma_j e^{-i(E_i + E_j)t} + \text{h.c.} \right), \tag{30}$$

and the Lindblad operators of the system are

$$C_n(\omega) = -2 \sum_{\omega = E_j - E_i} A_{i,j,n} \gamma_i^\dagger \gamma_j - 2 \sum_{\omega = E_j + E_i} B_{i,j,n} \gamma_i \gamma_j - 2 \sum_{\omega = -E_j - E_i} B_{i,j,n} \gamma_j^\dagger \gamma_i^\dagger. \tag{31}$$

The operators $C_n(\omega)$ rotate with frequency $\omega$ in the time evolution of the interaction picture such that the secular approximation can be used: one drops the fast rotating terms. This approximation can be only used when the energy differences in the system are smaller than the relaxation rates of the open system. Notice that, since we assume finite $N$, this approximation is valid as the energy differences $E_i - E_j$ of the discrete spectrum remains finite (excluding the zero modes). However one has to treat the zero modes carefully since their energy (for finite $N$) becomes exponentially small (nearly degenerate). This implies that, in the case of the topological regime, one can treat the zero modes as effectively degenerate with the ground state in the Lindblad equation, i.e. in Eq. 31 the Lindblad operator with $\omega = 0$ has terms in the sum with $E_j = E_i = 0$ for $i = 0_1$ and $j = 0_2$. Ultimately, this ensures the validity of the Lindblad equation for extremely long times in which the interaction picture of the zero modes can be assumed as time-independent $(\exp(-i(E_{0_i} + E_{0_j})t) \approx 1)$, see Eq. 30. The secular approximation becomes inaccurate near the critical points, where we have the transition between the finite energy excitation to the zero energy mode. Thus in the following analysis we exclude the

regime near the critical points. In the interaction picture (neglecting the Lamb shift terms), the final Lindblad equation takes the canonical form which reads

$$\frac{d\rho_s}{dt} = -\sum_\omega \frac{\kappa(\omega)}{2} \sum_n \mathcal{L}_n[\rho_s], \quad \mathcal{L}_n[\rho_s] = \left\{ C_n^\dagger(\omega)C_n(\omega), \rho_s \right\} - 2C_n(\omega)\rho_s C_n^\dagger(\omega). \quad (32)$$

Inserting Eq. (31) into Eq. (32), one finds the explicit form of the Lindblad equation which is reported in the appendix C. Hereafter we assume Ohmic dissipation for the transverse correlation function

$$\kappa(\omega) = \eta |\omega| \left[ \theta(\omega)(1 + n_B(\omega)) + \theta(-\omega)n_B(|\omega|) \right], \quad (33)$$

with $\eta$ setting the dissipative coupling strength to the environment and the bosonic function $n_B = 1/(e^{\omega/\Theta} - 1)$ with $\Theta$ the temperature. Notice that, in the limit of vanishing frequency and fixed temperature, one has formally $1/T_\phi = 2\eta\Theta$. In a finite size system, the average energy spacing in the single particle spectrum $|E_i - E_j|$ scales algebraically with the length $N$. By contrast, the separation between the energy of zero-energy excitations in the topological phase and the ground state $|GS\rangle$ scales exponentially with the length. At such vanishing energy differences, the presence of other source of noise beyond the Ohmic one can become important. Therefore, to take into account this effect, we use a phenomenological approach and we set the "zero frequency" damping rate $1/T_\phi$ as an independent parameter.

The derivation of the Lindblad equation as given by Eq. (32) (see also appendix C) is one of the main result of this work. This can be applied to any spin chain system which can be diagonalized via the Jordan-Wigner transformation. In the next section we discuss some applications for two specific cases: the transverse Ising model with winding number $g = 1$ in the topological phase and the extended model with $J_x^{(2)} > 0$ and winding number $g = 2$ in the topological phase.

# 4 Results for the transverse Ising model

We recover the transverse Ising model by setting the parameter $J_x^{(2)} = J_y^{(1)} = J_y^{(2)} = 0$ in the general class of the extended chain Hamiltonians. We focus on the low temperature limit $\Theta \ll E_{\text{gap}}$ where the gap is defined as $E_{\text{gap}} = E_0$ in the trivial regime and as $E_{\text{gap}} = \min_i E_i$ with $i \neq 0$ in the topological regime. Then the occupation of excited states is small and we can restrict to the lowest excitations of the spectrum formed by single or double particle excitation, as schematically shown in Fig. 6.

Hereafter we focus on the decoherence rate for an initial state given by the superposition between the even ground state $|GS\rangle$ and the (odd) lowest excitation $|0\rangle$, which is degenerate to the ground state $|GS\rangle$ in the topological regime. Solving the Lindblad equation we find:

$$\Gamma_{dec} = \frac{\Lambda_g^{(Ising)}}{T_\phi} + \Gamma_s + \Gamma_d, \quad (34)$$

where the first term arises from the fluctuations of the energy levels and corresponds to "pure dephasing". This is proportional to the overlap of the two wave functions of the zero mode and reads

$$\Lambda_g^{(Ising)} = \sum_n |\psi_{0,n}^R|^2 |\psi_{0,n}^L|^2. \quad (35)$$

The second term in Eq. (34) is associated to thermal fluctuations between the state $|0\rangle = \gamma_0^\dagger |GS\rangle$ and the single particle excitations $|i\rangle = \gamma_i^\dagger |GS\rangle$ and reads

$$\Gamma_s = 2 \sum_{i \neq 0} \sum_n A_{0,i,n} A_{i,0,n} \kappa(E_0 - E_i), \quad (36)$$

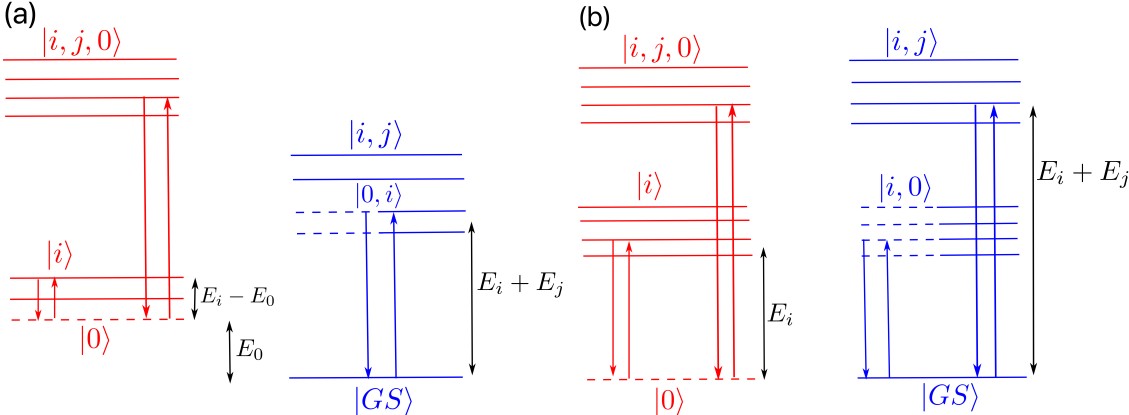

Figure 6: (a) Schematic view of the spectrum of the open transverse Ising chain in the trivial regime $J_x^{(1)} < B$. Energy transitions can occur between the even ground state $|GS\rangle$ and the two particle excitations with energy difference $E_i + E_j$ where $E_i$ or $E_j$ can be also $E_0$. Energy transitions are also possible between the lowest odd parity state $|0\rangle$ and the single particle excitations with smaller energy difference $E_i - E_0$ or between the single particle excitations and many particle excitations with higher energy difference $E_i + E_j$. (b) Schematic view of the spectrum in the transverse Ising open chain in the topological regime $J_x^{(1)} > B$ with the zero-energy mode $E_0 \to 0$. Energy transitions can occur between the even ground state $|GS\rangle$ and the two particle excitations at $E_i + E_j$ with $E_i, E_j \neq E_0$ or at energy $E_i + E_0 \approx E_i$. Energy transitions are also possible between the (almost) degenerate ground state $|0\rangle$ and the single particle excitation at energy $E_i - E_0 \approx E_i$ or with higher excitations at energy $E_i + E_j$.

with energy exchange $E_i - E_0$, see Fig. 6. Similarly, the third term in Eq. (34) is related to thermal fluctuations between the state $|GS\rangle$ and the double particle excitations $\gamma_i^\dagger \gamma_j^\dagger |GS\rangle = |i, j\rangle$ or the transitions between the state $|0\rangle$ and $|i, j, 0\rangle$ and energy difference $E_i + E_j$, see Fig. 6. The term $\Gamma_d$ reads

$$\Gamma_d = 2 \sum_{i \neq j} \sum_n B_{i,j,n} \left( B_{i,j,n} - B_{j,i,n} \right) \kappa(-E_i - E_j). \tag{37}$$

Notice that $|i, j\rangle$ also include the states with $i, j = 0$, namely $|i, 0\rangle$, see Fig. 6.

The contribution $\Gamma_d$ is exponentially small at temperature $\Theta \ll E_{\text{gap}}$. As shown in Fig. 6(a), $\Gamma_d$ connects transitions between the double particle excitations separated from the ground state by twice the gap energy $E_i + E_j \sim 2E_{\text{gap}}$ in the trivial phase. Thus the thermal energy is not sufficient to excite the lowest states to these higher excited states since $\Gamma_d \propto \exp(-2E_{\text{gap}}/\Theta)$. By contrast, $\Gamma_s$ can be relevant even at low temperature in the trivial regime since it involves transition with typical energy difference $E_i - E_0 \ll E_{\text{gap}}$, as shown in Fig. 6(a). Thus this interaction is not suppressed for $\Theta \ll E_{\text{gap}}$ and leads to additional decoherence in the trivial regime. In Fig. 7(a) we plot $\Gamma_s$ that rises with increasing $N$, since the energy difference between the state $|0\rangle$ to the next excitation $|i\rangle$ becomes smaller for larger $N$, see Fig. 6(a). Finally, the wave functions $\psi_{0,n}^L$ and $\psi_{0,n}^R$ does not correspond to localized states in the trivial regime and hence we have a finite overlap factor of order one

$$\Lambda_{g=0}^{(Ising)} = f_N^2 \sum_{n=1}^N \sin^2(k_0 n) \sin^2(k_0(N+1-n)). \tag{38}$$

This represent a finite pure dephasing contribution to the decoherence rate as plotted in Fig. 7(a).

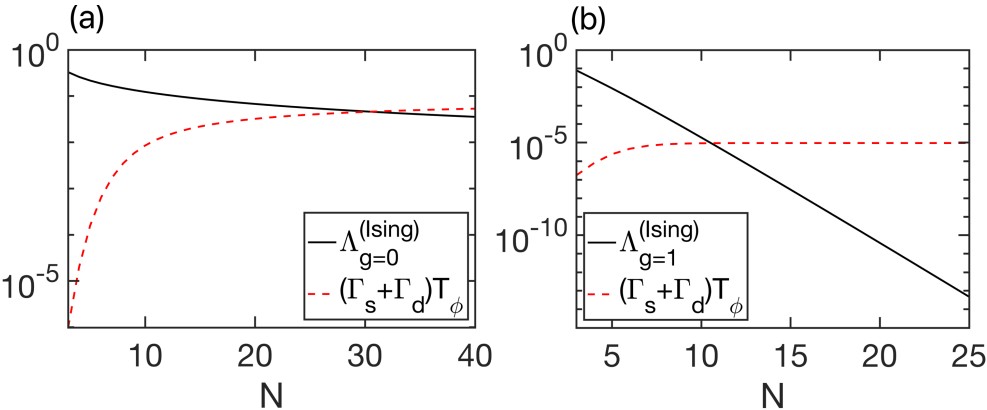

Figure 7: The different terms appearing in the dephasing rate $\Lambda_g^{(Ising)}$, $\Gamma_s$ and $\Gamma_d$ at $\Theta = 0.1 J_x^{(1)}$ as a function of $N$ for the open transverse Ising chain. (a) The trivial regime at $J_x^{(1)} = 0.5B$ and (b) the topological regime at $J_x^{(1)} = 2B$. Notice the different scale in the $y$ axis.

In the topological regime, $\Gamma_d$ has similar behavior as in the trivial regime hence is strongly suppressed as $\Gamma_d \approx \exp(-E_{\mathrm{gap}}/\Theta)$. Contrary to the previous trivial regime, the rate $\Gamma_s$ is also strongly suppressed by the gap $\Gamma_s \approx \exp(-E_{\mathrm{gap}}/\Theta)$ since it now involves transition between states $|i\rangle$ and the state $|0\rangle$, the latter almost degenerate with the ground state, namely $E_i - E_0 \sim E_i \sim E_{\mathrm{gap}}$ (see Fig. 6(b)). Finally, the pure dephasing contribution $\Lambda_g^{(Ising)}$ is directly related to the overlap of the localized states. In the large $N$ limit, it can be approximated as

$$\Lambda_{g=1}^{(Ising)} \approx N \big[ (J_x^{(1)}/B)^2 - 1 \big]^2 e^{-2q_0(N+1)}, \tag{39}$$

namely it is exponentially small due to localization of the end states $\psi_{0,n}^L$ and $\psi_{0,n}^R$. In Fig. 7(b) we plot the pure dephasing term $\Lambda_{g=1}^{(Ising)}$ and the sum of the two contributions $\Gamma_s + \Gamma_d$. Notice the different scale compared to the trivial regime. As expected $\Lambda_{g=1}^{(Ising)}$ has a strong dependence on $N$ whereas $\Gamma_s + \Gamma_d$ are almost constant as varying $N$ since their behavior is ruled by the presence of the energy gap.

## 5   Results for the extended model $J_x^{(2)} > 0$

We discuss now the extended model at finite value $J_x^{(2)} > 0$, as reported in Fig. 3(b),Fig. 4(b) and Fig. 5(b). We focus only on the topological regime with winding number $g = 2$ and in the zero temperature limit, namely $E_{\mathrm{gap}} \gg \Theta$. In this case the system has a fourfold degenerate ground state separated by a gap of order $E_{\mathrm{gap}} \sim 2(J_x^{(2)} - J_x^{(1)} - B)$.

By the analysis of the previous findings, we can restrict the dephasing dynamics in the ground state subspace, as the contribution due to the interaction with excitations scales with $\exp(-E_{\mathrm{gap}}/\Theta)$ and can be neglected in the zero temperature limit. This means the Lindblad operator can be expressed as

$$\hat{L}(\rho_s) = -\frac{1}{2T_\phi} \sum_n \big[ \{ \mathcal{P}_0 \sigma_n^z \mathcal{P}_0 \sigma_n^z \mathcal{P}_0, \rho_s \} - 2 \mathcal{P}_0 \sigma_n^z \mathcal{P}_0 \rho_s \mathcal{P}_0 \sigma_n^z \mathcal{P}_0 \big]. \tag{40}$$

with $\mathcal{P}_0$ the projector onto the ground state subspace.

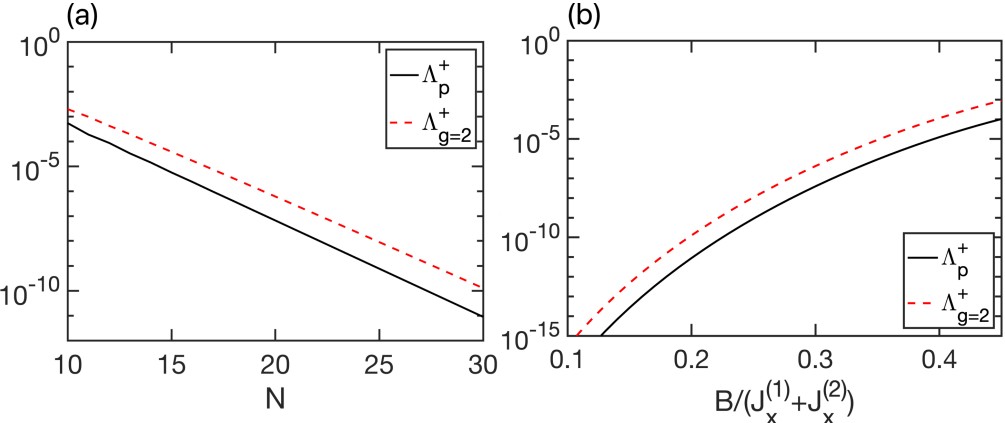

Figure 8: Coefficients $\Lambda_p^+$ and $\Lambda_{g=2}^+$ setting the dephasing dynamics (even parity subspace) in the open extended Ising model for $J_y^{(1)} = J_y^{(2)} = 0$ and $J_x^{(1)} = 0.25 J_x^{(2)}$ at (a) with $B = J_x^{(1)}$ and at (b) with $N = 30$ spins, in the topological regime with $g = 2$.

In the even subspace we set the populations as $p_g = \langle GS| \rho |GS\rangle$ and $p_0 = \langle 0_1, 0_2| \rho |0_1, 0_2\rangle$ and the off-diagonal coherent factor $\rho_{g,0} = \langle 0_1, 0_2| \rho |GS\rangle$. Then we obtain the equation

$$\frac{d(p_g - p_0)}{dt} = -\frac{\Lambda_p^+}{T_\phi} (p_g - p_0), \tag{41}$$

with

$$\Lambda_p^+ = \sum_n (\psi_{0_1,n}^L \psi_{0_2,n}^R - \psi_{0_1,n}^R \psi_{0_2,n}^L)^2, \tag{42}$$

and for the coherence factor we have

$$\frac{d\rho_{g,0}}{dt} = -\frac{\Lambda_{g=2}^+}{T_\phi} \rho_{g,0} - \frac{\tilde{\Lambda}_p^+}{T_\phi} (p_0 - p_g), \tag{43}$$

with

$$\Lambda_{g=2}^+ = \sum_n (\psi_{0_1,n}^L \psi_{0_1,n}^R + \psi_{0_2,n}^L \psi_{0_2,n}^R)^2, \tag{44}$$

and

$$\tilde{\Lambda}_p^+ = \sum_n \left(\psi_{0_1,n}^L \psi_{0_1,n}^R + \psi_{0_2,n}^L \psi_{0_2,n}^R\right)\left(\psi_{0_1,n}^L \psi_{0_2,n}^R - \psi_{0_2,n}^L \psi_{0_1,n}^R\right). \tag{45}$$

Examples of the behavior of the coefficients $\Lambda_p^+$ and $\Lambda_{g=2}^+$ appearing in time scales of the single qubit dephasing time are reported Fig. 8. Similar expressions are given for the odd subspace in the appendix B. The coefficients $\Lambda_p^+, \tilde{\Lambda}_p^+$ and $\Lambda_{g=2}^+$ are related to the overlap of the Majorana zero modes and they reduce exponentially with the length of the chain $N$, and with the decay length of the Majorana zero modes Eq. (21). In other words, the dephasing rate of the topological states is exponentially suppressed compared to the dephasing rate of an individual spin $1/T_\phi$. Notice, however, that in the limit $t \to \infty$, the steady state solution of Eq. (41) and Eq. (43) is simply $p_0 = p_g$ and $\rho_{g,0} = 0$.

In the last part we analyze the relaxation dynamics of the excited subspace. To simplify the notation, we discuss the relaxation in the even subspace and vanishing temperature limit. We set the populations $p_{j,0_1}$ as the occupation of the excited states in which the excitation

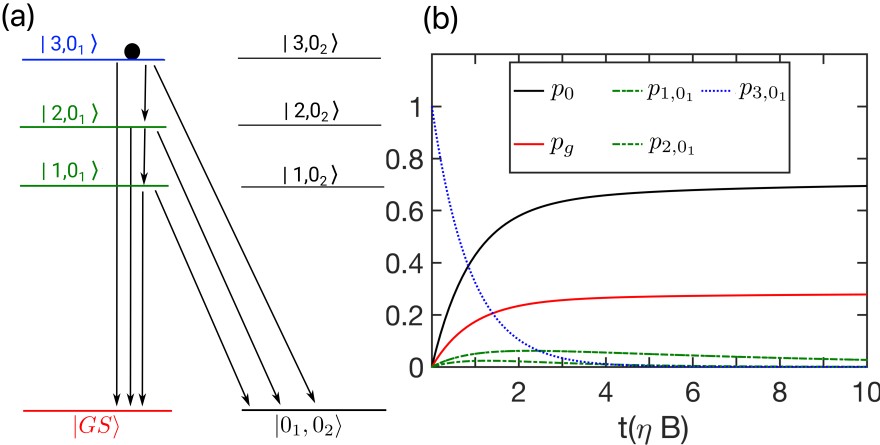

Figure 9: (a) Schematic decay paths in the even parity subspace in an open chain for the initial even state $|3, 0_1\rangle = \gamma_3^\dagger \gamma_{0_1}^\dagger |GS\rangle$ to one of the even ground states, $|GS\rangle$ or $|0_1, 0_2\rangle$ (or towards lower energy states $E_j < E_3$). (b) The populations for the different states in the even parity subspace as a function of time for $J_x^{(2)} = 4J_x^{(1)} = 8B$ for $N = 40$ spins.

$j$ and the zero energy mode $0_1$ are occupied. Notice that such states have energy $E_j$ in the topological regime. The populations $\{p_{j,0_1}\}$ satisfy the following set of coupled rate equations

$$\frac{dp_{j,0_1}(t)}{dt} \simeq -\left(W_{(j,0_1)\to g} + W_{(j,0_1)\to 0}\right) p_{j,0_1}(t) - \left(\sum_{E_{j'}<E_j} W_{j\to j'}\right) p_{j,0_1}(t) + \sum_{E_{j'}>E_j} W_{j'\to j}\, p_{j',0_1}(t),$$

(46)

where the rates are given by

$$W_{j\to j'} = \kappa(E_j - E_{j'})\, \chi^+_{(j,j')}, \tag{47}$$

$$W_{(j,0_1)\to g} = \kappa(E_j)\, \chi^-_{(j,0_1)}, \tag{48}$$

$$W_{(j,0_1)\to 0} = \kappa(E_j)\, \chi^+_{(j,0_1)}, \tag{49}$$

and the overlapping factor reads

$$\chi^\pm_{(j,j')} = \sum_n \left(\psi^R_{j,n}\psi^L_{j',n} \pm \psi^R_{j',n}\psi^L_{j,n}\right)^2. \tag{50}$$

The Eq. (46) describes the relaxation dynamics of the excited states $|j, 0_1\rangle$ which can decay directly towards one of the two ground states $|GS\rangle$ or $|0_1, 0_2\rangle$ or towards one excited state of lesser energy $E_{j'} < E_j$ (see Fig. 9). The last term in Eq. (46) is the positive ingoing flux due to the decay of states at energy $E_{j'} > E_j$. Eq. (46) is valid in the time scale in which we neglect internal relaxation in the ground state subspace. This is possible since we have a separation of the time scales: the prefactor $\kappa(E_j)$ in $W_{(j,0_1)\to g}$ and $W_{(j,0_1)\to 0}$ is ruled by the gap $\kappa(E_j) \simeq \kappa(E_{\mathrm{gap}})$ and, at the same time, the overlap factor $\chi^\pm_{(j,0_1)}$ involves a delocalized, extended state in the chain with one localized state at the end. The overlap factor $\chi^+_{(j,j')}$ involves two delocalized states. In other words, there is no exponential suppression of the rate as in the case of the internal dephasing in the ground state subspace. We solved numerically Eq. (46) to obtain the population $p_{j,0_1}(t)$ with the initial condition $p_{j,0_1}(0) = \delta_{ij}$, see Fig. 9.

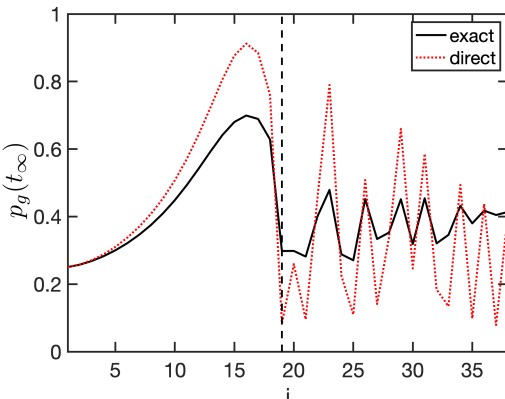

Figure 10: Exact final occupation $p_g(t_\infty)$ and the occupation $p_g^{(dir.)}$ (see text) of one of the two ground states in the even parity subspace as a function of the initial excited state $|i, 0_1\rangle$ (even parity) with $J_x^{(2)} = 4J_x^{(1)} = 8B$ and $N = 40$ (see Fig. 9). The vertical dashed line corresponds to the energy $E'_{\text{gap}}$ (see text and Fig.4(b)).

To complete the description of the relaxation dynamics, we have to write the equations for the populations of the two ground states $p_g$ and $p_0$

$$p_g(t) = \sum_j W_{(j,0_1)\to g} \int_0^t dt' \, p_{j,0_1}(t') \,, \tag{51}$$

$$p_0(t) = \sum_j W_{(j,0_1)\to 0} \int_0^t dt' \, p_{j,0_1}(t'). \tag{52}$$

One can check that, at long time $t \sim t_\infty$ with $t_\infty \gg 1/W$ but still $t_\infty \ll T_\phi/\Lambda_p^+$, the occupations saturate at values $p_g(t_\infty) \neq p_0(t_\infty)$ (with $p_g(t_\infty) = 1 - p_0(t_\infty)$) whereas for times $t > T_\phi/\Lambda_p^+$ the occupations of the ground states approach the values $p_0 = p_g = 1/2$. In Fig. 10 we report $p_g(t_\infty)$ for different initial states $(i, 0_1)$ of energy $E_i$. The difference between $p_g$ and $p_0$ strongly depends on the initial state for two reasons: (i) different (internal) decay paths towards lower lying excitations $E_j < E_i$, (ii) the different overlap with the two ground states. Naturally, higher excited states have more possible ways to decay towards more lower energy excitations which can become relevant and comparable, a priori, to the direct decay channel towards one of the two ground states. To distinguish between the two different mechanisms of dependence on the initial state, we compare the full expression Eq. (51) with the formula $p_g^{(dir.)} = W_{(i,0_1)\to g} \int_0^{t_\infty} dt' \, p_{i,0_1}(t')$ which contains only the direct decay from the initial state toward the ground state, see Fig. 10. We observe that the qualitative behavior of $p_g(t_\infty)$ is well reproduced by the $p_g^{(dir.)}$ with larger deviations as increasing the energy of the initial state. In particular, the direct decay description captures the different qualitative behavior at low and high energy.

In Fig. 10 $p_g(t_\infty)$ shows a regular behavior as a function of the initial state up to a maximum initial energy $E'_{\text{gap}}$ after which there are oscillations as increasing the energy of the initial state. This behavior can be explained by observing the single particle spectrum reported in Fig. 4(b): in the topological regime $g = 2$ ($B < J_x^{(2)} - J_x^{(1)}$) the spectrum has a regular energy spacing up to some energy $E'_{\text{gap}}$, reported as dashed line in Fig. 4(b). Above this energy a intersection of two different bundles of excitations appears. As a result of this intersection the wave functions of the excited states have alternating large and small overlaps with the first ground state $|GS\rangle$, leading to the oscillatory behavior in Fig.10 for $E_i > E'_{\text{gap}}$. This energy $E'_{\text{gap}}$

plays the role, roughly speaking, of an effective, secondary gap of the system, which closes at the transition for $g = 1 \to g = 0$ (whereas the primary gap with energy $E_{\mathrm{gap}}$ closes at the transition $g = 2 \to g = 1$). Hence the topology of the model is not only a ground state property but can also appear in the relaxation dynamics involving (low energy) excited states.

# 6 Summary

In order to understand the robustness of the topological properties of spin chains affected by dissipative interaction with the environment, we studied an extended quantum Ising model in which each single spin is subject to a longitudinal dissipative interaction with a local bath.

For the ground state subspace, we derive the formula for the dephasing rates, in a given parity subspace, that incorporate the two Majorana zero modes showing the robustness of the ground state subspace against the transverse dissipative coupling. The inclusion of a more general dissipative coupling (i.e. with longitudinal) is a interesting future perspective. Furthermore we have shown that the topology can also influence the relaxation dynamics of excited states. Here the secondary gap (which appears for $g = 2$) determines the relaxation behavior of the excited states and the resulting occupation imbalance of the ground states. It would be interesting if similar behavior can be observed in other topological systems.

Although we focus on the extended Ising chain with a specific form of the next nearest neighbor interaction, with winding number $g = 2$ in the topological phase, our results can be readily extrapolated to understand the relaxation and dephasing dynamics of the whole extended class of quantum Ising chains.

## Acknowledgements

This work was supported by the German Excellence Initiative through the Zukunftskolleg, the Deutsche Forschung Gemeinschaft (DFG) through the SFB 767, Project No. 32152442, and by the MWK Baden-Württemberg Research Seed Capital (RiSC) funding.

## A Diagonalization of the Ising chains

Rewriting the fermionic Hamiltonian in Eq. (6) in matrix representation

$$H_c = \frac{1}{2} \sum_{n,m} \begin{pmatrix} c_n^\dagger & c_n \end{pmatrix} \begin{pmatrix} t_{nm} & \Delta_{nm} \\ \Delta_{mn} & -t_{nm} \end{pmatrix} \begin{pmatrix} c_m \\ c_m^\dagger \end{pmatrix}, \tag{53}$$

with

$$t_{nm} = \delta_{n,m} 2B - \delta_{n+1,m} \left( J_x^{(1)} + J_y^{(1)} \right) - \delta_{n+2,m} \left( J_x^{(2)} + J_y^{(2)} \right), \tag{54}$$

and

$$\Delta_{nm} = -\delta_{n+1,m} \left( J_x^{(1)} - J_y^{(1)} \right) - \delta_{n+2,m} \left( J_x^{(2)} - J_y^{(2)} \right). \tag{55}$$

Inserting the transformation of Eq. (15) into the Hamiltonian Eq. (53), we impose that such transformation diagonalizes the Hamiltonian in the form of Eq. (14) and we get the equations

$$E_i \left( \psi_{i,n}^L + \psi_{i,n}^R \right) = \sum_m \left( t_{nm} \left( \psi_{i,m}^L + \psi_{i,m}^R \right) + \Delta_{nm} \left( \psi_{i,m}^L - \psi_{i,m}^R \right) \right), \tag{56}$$

and

$$E_i\left(\psi_{i,n}^L - \psi_{i,n}^R\right) = \sum_m \left(t_{nm}\left(\psi_{i,m}^L - \psi_{i,m}^R\right) - \Delta_{nm}\left(\psi_{i,m}^L + \psi_{i,m}^R\right)\right). \tag{57}$$

Setting the matrix $(\bar{T})_{nm} = t_{nm}$ and $(\bar{\Delta})_{nm} = \Delta_{nm}$, and the vectors $(\vec{\psi}_i^{L,R})_n = \psi_{i,n}^{L,R}$ the two Eqs. (56,57) are represented in the following matrix form

$$E_i\,\vec{\psi}_i^{\,L} = (\bar{T} - \bar{\Delta})\vec{\psi}_i^{\,R}, \tag{58}$$

$$E_i\,\vec{\psi}_i^{\,R} = (\bar{T} + \bar{\Delta})\vec{\psi}_i^{\,L}. \tag{59}$$

In general case, we have solved numerically the last equations to find the eigenvectors and the respective eigenvalues $E_i$ (single particle energy spectrum).

## B  Dephasing dynamics in the extended model for the odd subspace

We set the population $p_{0_1} = \langle 0_1| \rho |0_1\rangle$ and $p_{0_2} = \langle 0_2| \rho |0_2\rangle$ and the off-diagonal (coherent) factor $\rho_{0_1 0_2} = \langle 0_1| \rho |0_2\rangle$. Then we derive the following equations

$$\frac{d(p_{0_2} - p_{0_1})}{dt} = -\frac{\Lambda_p^-}{T_\phi}(p_{0_2} - p_{0_1}), \tag{60}$$

with

$$\Lambda_p^- = \sum_n \left(\psi_{0_1,n}^R \psi_{0_2,n}^L + \psi_{0_2,n}^R \psi_{0_1,n}^L\right)^2. \tag{61}$$

The second equation for $\rho_{0_1 0_2}$ reads

$$\frac{d\rho_{0_1 0_2}}{dt} = -\frac{\Lambda_{g=2}^-}{T_\phi}\rho_{0_1 0_2} - \frac{\tilde{\Lambda}_p^-}{T_\phi}(p_{0_2} - p_{0_1}), \tag{62}$$

with

$$\Lambda_{g=2}^- = \sum_n (\psi_{0_1}^L \psi_{0_1}^R - \psi_{0_2}^L \psi_{0_2}^R)^2, \tag{63}$$

and

$$\tilde{\Lambda}_p^- = \sum_n \left(\psi_{0_2,n}^L \psi_{0_2,n}^R - \psi_{0_1,n}^L \psi_{0_1,n}^R\right)\left(\psi_{0_1,n}^L \psi_{0_2,n}^R + \psi_{0_2,n}^L \psi_{0_1,n}^R\right). \tag{64}$$

## C  Explicit formula of the Lindblad equation

One can derive the full Lindblad equation by using the spectral representation as given by Eq. (31). The complete form of the Lindblad equation for the open chain reads

$$
\begin{aligned}
\frac{d\rho_s}{dt} = \sum_n \Bigg( &4 \sum_{E_i - E_j = E_l - E_k} A_{i,j,n} A_{k,l,n} \kappa(E_i - E_j)\left(\gamma_k^\dagger \gamma_l \rho_s \gamma_i^\dagger \gamma_j - \frac{1}{2}\{\gamma_i^\dagger \gamma_j \gamma_k^\dagger \gamma_l, \rho_s\}\right) \\
&+ 4 \sum_{E_i - E_j = E_{ik} + E_l} A_{i,j,n} B_{l,k,n} \kappa(E_i - E_j)\left(\gamma_k \gamma_l \rho_s \gamma_i^\dagger \gamma_j - \frac{1}{2}\{\gamma_i^\dagger \gamma_j \gamma_k \gamma_l, \rho_s\}\right) \\
&+ 4 \sum_{E_i - E_j = -E_k - E_l} A_{i,j,n} B_{l,k,n} \kappa(E_i - E_j)\left(\gamma_l^\dagger \gamma_k^\dagger \rho_s \gamma_i^\dagger \gamma_j - \frac{1}{2}\{\gamma_i^\dagger \gamma_j \gamma_l^\dagger \gamma_k^\dagger, \rho_s\}\right) \\
&+ 4 \sum_{E_i + E_j = E_k - E_l} B_{j,i,n} A_{k,l,n} \kappa(-E_i - E_j)\left(\gamma_k^\dagger \gamma_l \rho_s \gamma_i \gamma_j - \frac{1}{2}\{\gamma_i \gamma_j \gamma_k^\dagger \gamma_l, \rho_s\}\right) \\
&+ 4 \sum_{E_i + E_j = E_k + E_l} B_{j,i,n} B_{l,k,n} \kappa(-E_i - E_j)\left(\gamma_l^\dagger \gamma_k^\dagger \rho_s \gamma_i \gamma_j - \frac{1}{2}\{\gamma_i \gamma_j \gamma_l^\dagger \gamma_k^\dagger, \rho_s\}\right) \\
&+ 4 \sum_{E_i + E_j = E_k + E_l} B_{j,i,n} B_{l,k,n} \kappa(E_i + E_j)\left(\gamma_k \gamma_l \rho_s \gamma_j^\dagger \gamma_i^\dagger - \frac{1}{2}\{\gamma_j^\dagger \gamma_i^\dagger \gamma_k \gamma_l, \rho_s\}\right) \\
&+ 4 \sum_{E_i + E_j = E_l - E_k} B_{j,i,n} A_{k,l,n} \kappa(E_i + E_j)\left(\gamma_k^\dagger \gamma_l \rho_s \gamma_j^\dagger \gamma_i^\dagger - \frac{1}{2}\{\gamma_j^\dagger \gamma_i^\dagger \gamma_k^\dagger \gamma_l, \rho_s\}\right) \\
&+ 4 \sum_{E_i + E_j = -E_l - E_k} B_{j,i,n} B_{l,k,n} \kappa(E_i + E_j)\left(\gamma_l^\dagger \gamma_k^\dagger \rho_s \gamma_j^\dagger \gamma_i^\dagger - \frac{1}{2}\{\gamma_j^\dagger \gamma_i^\dagger \gamma_l^\dagger \gamma_k^\dagger, \rho_s\}\right) \\
&+ 4 \sum_{E_i + E_j = -E_l - E_k} B_{j,i,n} B_{l,k,n} \kappa(-E_i - E_j)\left(\gamma_k \gamma_l \rho_s \gamma_i \gamma_j - \frac{1}{2}\{\gamma_j \gamma_i \gamma_k \gamma_l, \rho_s\}\right) \Bigg).
\end{aligned}
\tag{65}
$$

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
