# Peer review of "Decoherence and relaxation of topological states in extended quantum Ising models"

_SciPost Physics, doi:SciPost Phys. 6, 037 (2019)_

## Round 1 · Referee Report · Anonymous · 2019-1-14

Strengths

1 - important topic
2 - good structure
3 - interesting analysis

Weaknesses

1- some points seem to use too much jargon

Report

I have read with interest the work by Weisbrich et al.
I think that it is a good paper.
It concerns a very important topic of topological systems in contact with an environment.
The authors use standard tools from quantum optics, however it would be good to comment on their validity, especially in presence of small energy differences.
The structure of the paper is very good, however in some parts the authors use terms like T_1 and T_2 which are not clear to a broad audience (see requested changes).
A comparison to the paper by Huang et al (see requested changes), could set this paper into context better.

Requested changes

1 - The authors could emphasize they can consider different local baths could be.
2 - Scipost is for a general audience, so I would suggest that the authors spend a line or two in explaining what T_1 and T_2 are. Especially since this is discuss early in the paper, but it becomes really clear only in section 3.
3 - I did not find it very clear that the authors were considering both open and periodic boundary conditions. This could be reinforced, for example, in the captions of the figures.
4 - It is also not very clear to me in which parity sector in most of the figures.
5 - I am confused by (9). We have an H^\pm_S on the right-hand side, but on the left-hand side there is no \pm.
6 - One important aspect of topological systems is the presence of the zero-energy modes. This also implies the presence of degeneracies. In this case the authors state that there are no degeneracies due to finite-size effects. However, the master equation used by the authors is a weak coupling equation, and hence it suffers in presence of small energy differences, requiring the use of very small couplings. I would deem important that the authors at least discuss the ideal case, in which there are degeneracies, and the actual case, in which there are very small gaps.
7 - the use of T both for temperature and time scale (although in the second case with a sub-index) is a bit confusing. Maybe use capital \Theta for temperature?
8 - I have recently studied another interesting and very relevant paper by Huang et al. "Dissipative Majorana quantum wires" arXiv:1812.04471
It seems to me that this paper is very relevant. I would suggest the authors not only to cite it, but also to comment on their findings but also to compare them to theirs.
9 - do the authors have a deeper understanding of the change in the occupation of the modes shown in Fig.10? The exact one seems to follow an Airy function profile, but independent of what the shape of the function really is, it would be good to provide some understanding.
10 - I find that the summary could be extended. It is too coincise to have a good overview of the paper. In particular, the sentence "In each of the two subspace..." until "towards the degenerate ground state" is not clear to me.

After this changes have been considered, I recommend the paper for publication.

  • validity: good
  • significance: good
  • originality: high
  • clarity: high
  • formatting: excellent
  • grammar: excellent

Author:  Hannes Weisbrich  on 2019-03-13  [id 465]

(in reply to Report 1 on 2019-01-14)

We thank the referee for his/her positive useful comments and questions. Below we provide the answers to the questions that she/he raised. We think that we improved the readability of manuscript following the suggestion of the Referee.

>> The authors use standard tools from quantum optics, however it would
>> be good to comment on their validity, especially in presence of small
>> energy differences.
and related
>> One important aspect of topological systems is the presence of the zero-
>> energy modes. This also implies the presence of degeneracies. In this case
>> the authors state that there are no degeneracies due to finite-size effects.
>> However, the master equation used by the authors is a weak coupling
>> equation, and hence it suffers in presence of small energy differences,
>> requiring the use of very small couplings. I would deem important that the
>> authors at least discuss the ideal case, in which there are degeneracies,
>> and the actual case, in which there are very small gaps.

The derivation of the Lindblad equation is based on one approximation known as secular approximation (see book of Breuer and Petruccione) which is based on the assumption that the energy difference of the discrete spectrum of the system are larger than relaxation rates.
By consequence, this secular approximation suffers in the case when the energy difference between the states vanishes. For this we have distinguish between three cases:

(1) the energy differences are much larger than the relaxation rate, i.e. the trivial regime
in our case of the model (no degeneracy) or the regime of short chains

(2) the energy difference is exponentially small, i.e. the states are
effectively degenerate. In this case we didn’t use the secular approximation but we
treated the two lowest energy states as degenerate - the two ground states.
On the other hand, we assumed the other finite energy states as non degenerated.

(3) the energy difference are comparable to the relaxation rate, i.e. in our problem this
would be the case near the critical points, where the zero modes energy changes from finite to (nearly) degenerate with the ground state, or when we address the cross-over from short to long chains in the extended model.

The last regime was beyond the scope of our analysis. Anyway, we considered the difference between the case 1) and 2) into account. This difference is automatically taken into account in the expression of the Lindblad equation in which a part of the Lindblad operator is nonzero when the states are treated as “degenerate” in the sense explained above.

We explained better our analysis in the new version of the manuscript on page 11 after Eq.31.

>> The structure of the paper is very good, however in some parts the authors
>> use terms like T_1 and T_2 which are not clear to a broad audience
>> (see requested changes).
and related
>> Scipost is for a general audience, so I would suggest that the authors
>> spend a line or two in explaining what T_1 and T_2 are. Especially since this >> is discuss early in the paper, but it becomes really clear only in section 3.

We modified the part in the introduction in way to define the physical meaning of these two time constants. We think that the text is now accessible for non-experts.

We added a sentence of four lines in the introduction at the beginning on page 3.

>> A comparison to the paper by Huang et al (see requested changes),
>> could set this paper into context better.
and related
>> I have recently studied another interesting and very relevant paper by
>> Huang et al. "Dissipative Majorana quantum wires" arXiv:1812.04471.
>> It seems to me that this paper is very relevant. I would suggest the authors >> not only to cite it, but also to comment on their findings but also to
>> compare them to theirs.

This paper was indeed interesting and relevant for us. In this work the authors considered the local "density” operator coupled to the bath, which means that, in the spin Hamiltonian representation, the spins are coupled to the bath via the same spin component parallel to the external magnetic field of the Ising model, as in our case.

However, in the work arXiv:1812.04471, the authors addressed the effect of such dissipation on the phase diagram, which requires to go beyond the Lindblad-Markovian approach and to use the Path Integral approach.
By contrast, in our work, we have studied this kind of dissipation but in the Lindblad-Markovian approach, since we were interested in the relaxation/decoherence dynamics and not in the phase diagram. Moreover, we have also generalized our analysis to extended Ising models characterized by more than one Majorana mode.

We cited this paper in our introduction of the new version.

>> do the authors have a deeper understanding of the change in the
>> occupation of the modes shown in Fig.10? The exact one seems to follow
>> an Airy function profile, but independent of what the shape of the function
>> really is, it would be good to provide some understanding.

We don’t have an analytical formula for the function describing the behavior of the occupation imbalance of the ground states in respect to an initial excited state. But we have a simple explanation which is based on the “crossing” of a second band shown in Fig. 4.
We realized that this part was not written in a proper way. We have substantially modified this part in order to make our argument clear.

We added a couple of sentence in the last paragraph of Section 5 on page 18.

>> I find that the summary could be extended. It is too coincise to have a good
>> overview of the paper. In particular, the sentence "In each of the two
>> subspace..." until "towards the degenerate ground state" is not clear to me.

We have modified the conclusions (Section 6) following the recommendation of the Referee.

We also added the other minor requested changes by the Referee.

---

## Round 1 · Referee Report · Jean-Sébastien Caux · 2019-1-28

Strengths

- An enjoyable read: focused, to-the-point, honest science;
- The subject is timely and interesting;
- The paper is very legible and clearly written;
- Nontrivial extended models with next nearest neighbour, 3-body interactions are treated;
- There is a good list of cited contextual references.

Weaknesses

No major weakness. The paper accomplishes what it sets out to do.

For appearance's sake, if I must be critical:
- some extra motivations for the choice of models could have been given;
- transverse couplings to the environment are neglected, but wouldn't these drastically affect the (long) timescales given for the transitions?
- though I liked the last couple of paragraphs before the summary, I was left a bit puzzled as to how stable the results are to the inclusion of additional interactions. I understand this is speculation, but an outlook (with perhaps a couple of challenges for further work) would have worked well at this point.

Report

Please see the list of changes below, which mixes pointing out simple typos with some bigger (but all easy to address) comments.

Provided these are addressed, I recommend the paper for acceptance.

Requested changes

Technicalities:
- references should be hyperref'ed

Typos:
Abstract:
"manyfold" -> "multidimentional" or something similar; "manyfold" means "many times" [this to be changed also in other places in the paper]

p2: when used as adjective: one-dimensional -> one-dimensional [also in other places]

p3: in presence of -> in the presence of

p3: cross-over -> crossover

p5: perhaps give a better motivation for the choice of correlated hopping term. If it's only chosen because it makes the fermionized Hamiltonian simple, then say this clearly. If there is a further physical motivation, then provide it.

p5: maybe worth defining $\nu_n$ separately for legibility

p5: in equation 6, second line, the index of the $J_{x-y}$ should be 2, not 1

p6: in equation 10, the last sign should be -, not +

p7: is referred as -> is referred to as

p8: equation 17, don't repeat "(transv. Ising)" on two lines

p8: after eqn 17, paragraph is not flush left

p8: Example of -> An example of [also p9]

p8: "This represent" -> This represents

p8: "Hence the system results to have" -> rephrase this whole sentence

p9: appearance of first -> appearance of a first

p3: after equation 20, in text: orthogonality condition should involve $\psi^* \psi$ instead of $\psi \psi$ (I know the approximate wavefunctions given are real, but in general this isn't the case)

p9: Before to discuss -> Before discussing

p10: one factorize -> one factorizes
the relevant quantity -> the relevant quantities

p10: "local bathes" -> local baths
"large scaled spin chain" -> long spin chain

p14: why does N start around 10 for the data in Fig. 7? Log scale better for the b) subplot in the topological regime?

p15: "solution of the Eq. (41)... " -> solution of Eq. (41) ...

p18: "of a effective" -> of an effective

p18: "in each of the two subspace" -> in each of the two subspaces

p20: beginning of Appendix C: "Lindlblad"

  • validity: high
  • significance: good
  • originality: good
  • clarity: high
  • formatting: excellent
  • grammar: good

Author:  Hannes Weisbrich  on 2019-03-13  [id 464]

(in reply to Report 2 by Jean-Sébastien Caux on 2019-01-28)
Category:
answer to question
correction

We thank Prof. Caux for the positive assessment of our work and for his time for
reading carefully our manuscript. Below we answered to the questions that he raised.

>> some extra motivations for the choice of models could have been given;

We studied the extended Ising model, beyond the standard transverse Ising model, since it host two-fold degenerate ground states of same parity (two Majorana modes).
The simple Ising open chain of finite length is characterized by two almost degenerate ground states with different parity in the topological phase (ferromagnetic). Therefore, it is impossible to connect them by slowly varying the parameters of the Hamiltonian, due to their different parity.
On the other hand, the extended Ising model opens the possibility to implement adiabatic quantum computations in the subspace of given parity.

We added a sentence of four lines in the text when we introduce the extended Ising model (beginning of page 4).

>> transverse couplings to the environment are neglected, but wouldn't these
>> drastically affect the (long) timescales given for the transitions?

We completely agree with this statement and this is indeed a natural question that we have also asked ourselves.

In fact, we start our analysis by assuming that the dissipative transverse interaction leads to a very low relaxation / dephasing (decoherence) rates, or slow time scales whereas the longitudinal dissipative interaction represents the leading effect. This analysis a priori is based on the results for the single qubit.

However, within our Lindblad-Markovian approach, the coupling strength of the interaction with the environment enters as a parameter multiplying the square modulus of matrix elements. Introducing the interaction, the eigenstates of the Hamiltonian changes as well as the matrix elements appearing in the rates.

We show that, in the topological regime, the relaxation and the dephasing rates associated to the longitudinal dissipative coupling becomes exponentially small (with the length) compared to the original rate (the dephasing of the single qubit). Indeed these rates can be (a-posteriori) comparable to rates associated with dissipative transverse interaction which we neglected. This is of course true if one considers realistic systems of qubits.

Our objective was to analyze the cross-over region (intermediate lengths of the chain), namely how the system approaches the topological region in which the decoherence is suppressed.
We didn’t aim to calculate the exact relaxation/dephasing rate for the general case which can be of interest for many experimental qubits systems. This would be an interesting topic for future works.
We added a couple of sentence at the conclusions for explaining better this point (Page 18).

>> though I liked the last couple of paragraphs before the summary, I was left a bit
>> puzzled as to how stable the results are to the inclusion of additional
>> interactions. I understand this is speculation, but an outlook (with perhaps
>> a couple of challenges for further work) would have worked well at this point.

Concerning the last result (the behavior of the occupation imbalance of the two ground states depending on the initial excited state) we don’t know exactly the effect of different interactions.
Our point here is the following:
Generally, the topological number appears as a ground state property. For instance, in the extended Ising models, the winding number is associated to the number of Majorana modes and the ground state degeneracy. In contrast, we have shown that topology can also appear in the relaxation dynamics involving (low energy) excited states.
In our case, this was explained by the crossing of two different “bands“ associated to the higher winding number g=2.
We don’t know how general this result is and if it occurs in other topological systems as well.
But, at least, we proved that, in one system, topological properties can also appear in the relaxation dynamics.

We added a couple of sentence at the end of Section 5 and in the outlook (Section 6) on page 18.

We also implemented the other minor requested changes by Prof. Caux.

---

## Editorial Decision

published